# Anthropogenic impacts on tidal creek sedimentation since 1900

Molly C. Bost[1,2,3]*, Charles D. Deaton[1], Antonio B. Rodriguez[1,4], Brent A. McKee[2], F. Joel Fodrie[1,4], Carson B. Miller[1]

**1** Institute of Marine Sciences, University of North Carolina at Chapel Hill, Morehead City, North Carolina, United States of America, **2** National Oceanic and Atmospheric Administration, National Centers for Coastal Ocean Science, Beaufort, North Carolina, United States of America, **3** CSS-Inc., Fairfax, Virginia, United States of America, **4** Department of Earth, Marine and Environmental Sciences, University of North Carolina at Chapel Hill, Chapel Hill, North Carolina, United States of America

\* mbost0@gmail.com

**Data Availability Statement:** All relevant data are within the paper and its Supporting information files.

## Abstract

Land cover and use around the margins of estuaries has shifted since 1950 at many sites in North America due to development pressures from higher population densities. Small coastal watersheds are ubiquitous along estuarine margins and most of this coastal land-cover change occurred in these tidal creek watersheds. A change in land cover could modify the contribution of sediments from tidal creek watersheds to downstream areas and affect estuarine habitats that rely on sediments to persist or are adversely impacted by sediment loading. The resilience of wetlands to accelerating relative sea-level rise depends, in part, on the supply of lithogenic sediment to support accretion and maintain elevation; however, subtidal habitats such as oyster reefs and seagrass beds are stressed under conditions of high turbidity and sedimentation. Here we compare sediment accumulation rates before and after 1950 using ²¹⁰Pb in 12 tidal creeks across two distinct regions in North Carolina, one region of low relief tidal-creek watersheds where land cover change since 1959 was dominated by fluctuations in forest, silviculture, and agriculture, and another region of relatively high relief tidal-creek watersheds where land-use change was dominated by increasing suburban development. At eight of the creeks, mass accumulation rates (g cm⁻² y⁻¹) measured at the outlet of the creeks increased contemporaneously with the largest shift in land cover, within the resolution of the land-cover data set (~5-years). All but two creek sites experienced a doubling or more in sediment accumulation rates (cm yr⁻¹) after 1950 and most sites experienced sediment accumulation rates that exceeded the rate of local relative sea-level rise, suggesting that there is an excess of sediment being delivered to these tidal creeks and that they may slowly be infilling. After 1950, land cover within one creek watershed changed little, as did mass accumulation rates at the coring location, and another creek coring site did not record an increase in mass accumulation rates at the creek outlet despite a massive increase in development in the watershed that included the construction of retention ponds. These abundant tidal-creek watersheds have little relief, area, and flow, but they are impacted by changes in land cover more, in terms of percent area, than their larger riverine counterparts, and down-stream areas are highly connected to their associated watersheds. This work expands the scientific understanding of connectivity between

**Funding:** FJF and ABR 2016-H-056 NC Division of Marine Fisheries https://deq.nc.gov/about/divisions/marine-fisheries/grant-programs/coastal-recreational-fishing-license-grant-program#funding-information Funders had no role in study design, data collection and analysis, decision to publish, or preparation of the manuscript.

**Competing interests:** The authors have declared that no competing interests exist.

lower coastal plain watersheds and estuaries and provides important information for coastal zone managers seeking to balance development pressures and environmental protections.

## Introduction

Drowned river-mouth estuaries primarily receive sediment from their formative rivers. A portion of that sediment is deposited in the estuary and forms the substrate on which ecologically and economically important habitats, such as sea grass, oyster reefs, mangroves, and salt marshes, colonize. Changes in sediment accumulation in the coastal zone, however, can result in either the proliferation or demise of those habitats and the quantity and type of ecosystem services provided [1, 2]. A shift toward more frequent and/or larger pulses of sediment delivered to an estuary can force habitats to transition, such as tidal flats transitioning to intertidal saltmarsh, or sandy subtidal flats transitioning to muddy subtidal flats [3]. If sedimentation is too high, seagrass and oyster reefs can become buried and converted to mud or sandflat environments [4]. A reduction in sediment supply can cause intertidal habitats like saltmarsh and mangrove to transition to open water due to excessive inundation with sea-level rise [5–9] and communities subsequently lose the protection those habitats provide from storms [10–13].

Humans have significantly modified river sediment load, discharge, and the degree to which watersheds are connected to the coast [14, 15]. Land-cover changes, such as clearing forests, commonly promote soil erosion and increased sediment load in rivers, with relief providing an important first-order control on the extent of landscape erosion and sediment transport [16]. Development of urban and suburban centers is associated with the channelization of catchment basins, increased flow velocities and increased connectivity between landscapes and downstream environments; however, reservoirs interrupt sediment transport across the coastal plain [17–19]. An inferred decrease in sediment delivery to estuaries, based on river gauges positioned ~15–100 km landward of river outlets, has been attributed to the construction of dams, which peaked in the United States around 1950 [20, 21]. This inference assumes land-cover changes in the lower coastal plain contributes little to estuarine sedimentation, and conflict with direct measures of increasing sediment accumulation rates in many North American estuaries [22] and a net global increase in deltaic land area over the past 30 years [23].

Coastal population has been rising for more than half a century and is a common proxy for land-use change [24]. Coastal county populations throughout the United States have increased by 39% since the 1970s and population density is over six times greater in coastal counties than inland ones [25]. Most of the population increase and associated change in land cover in coastal counties occurs seaward from where the main river discharges into the heads of estuaries. Those downstream areas are connected to estuaries via multiple smaller watersheds confined to the lower coastal plain. Those smaller watersheds are tidal creeks, which we define for this study as small channel networks that drain lower-coastal plain watersheds less than 50 km$^2$, are tidal along their entire length, discharge into larger estuaries, lagoons, or back-barrier sounds, and are fringed by saltmarsh complexes along their main stem. Tidal creeks are comprised of an upper reach where the channel is constricted and meanders along a fringing saltmarsh complex, and a lower embayed open-water region of confluence between the lower tidal-creek basin and the main estuary it drains into. In this open-water region, salt marsh islands, oyster reefs, seagrass beds, and mudflats are common.

The relief of tidal-creek watersheds and the area of the lower embayed region is mainly controlled by the type of incised valley they formed within, including coastal prism and tributary

incised valleys [26]. Coastal prism incised valleys formed across the break in slope between the low-gradient coastal plain and the steep shoreface of the previous highstand shoreline when sea level was lower. As sea level rose during the late Holocene that antecedent topography flooded, and the resulting tidal-creek watersheds are mainly confined to the high-relief topography (~5–20 m) of the previous highstand shoreline and embayed areas that scale with drainage basin size [26]. Tributary incised valleys formed along the margins of larger incised trunk valleys. The high convexity along the edge of the incised valley, between the steep valley flank and the flat coastal plain, increased stream power and promoted erosion and knickpoint migration when sea level was lower. That steep relic landscape morphology was mostly flooded during the late Holocene rise in sea level, and the resulting tidal-creek watersheds are confined to the low-relief coastal plain (~<5 m) and embayed areas are commonly larger than what would be predicted by their small drainage basin size [26].

The transfer of sediment from upstream to downstream areas in a river system (connectivity [27]) generally decreases as catchment area increases because of sediment storage in floodplains. Higher topographic gradients are typically correlated with an increase in upstream to downstream connectivity, and low relief coastal plain rivers are assumed to be disconnected from land cover changes in their watershed [28]; however, there are examples where that assumption does not apply [3, 29–31]. The smallest lower coastal plain watersheds are tidal creeks and their short reach and little to no floodplain could make them important sources of sediment to estuaries.

Sediment is supplied to a tidal creek from its drainage basin and from the estuary it drains into. Sediment moves primarily from upstream to downstream during rain events and sources include runoff, resuspension, and bank erosion. Tidal-creek watersheds have little relief, highly variable vegetation cover, and a wide range of anthropogenic impacts so the potential for erosion within the watershed can vary substantially across small spatial scales. Sediment moves from the estuary into the tidal creek during flood tide or storm surge and is sourced mainly from surrounding drainage basins, offshore, resuspension, and shoreline erosion.

Sediment deposition in the embayed zone of a tidal creek is driven by sediment supply and accommodation, which change over a variety of timescales. Accommodation is the amount of space available for sediments to accumulate in, which near the coast, is closely tied to the level of wave base and the rate of relative sea-level rise (RSLR; [32]). Over centennial timescales accommodation is mainly driven by RSLR and sediment flux by climate. Shorter timescales (annual to decadal) encompass changes in watershed land cover and storminess, which results in higher sediment flux to the tidal-creek embayment, and higher resuspension of the bed as wave base deepens, respectively. The water depth of the lower embayed portion of tidal creeks is not necessarily in equilibrium with storm wave base, as is common for larger estuaries and lagoons [33]. The tidal creek embayment within many tributary incised valleys has a surplus of sediment accommodation because they are oversized in comparison to the creek's watershed [34]. Tidal creeks in coastal prism incised valleys, however, are more accommodation limited because the size of the lower embayed portion scales with the size of the tidal-creek watershed and contains numerous intertidal patches of oyster reefs and salt marshes. The objective of this study was to test the hypothesis that tidal creek connectivity and contribution of sediment to downstream estuarine areas are low. We examined the effects of changes in land cover of tidal-creek watersheds in North Carolina, U.S.A. (NC), which vary by orders of magnitude in drainage basin size and relief, experience different tidal ranges, and have been impacted by different types of landscape change, on sediment accumulation in the lower embayment. Connectivity is measured as the time between when a major change in land cover occurs in the watershed and when sediment accumulation

increases in the downstream embayment. A low sediment contribution to downstream areas is recognized as sediment accumulation rates not keeping pace with RSLR. In our study area along the 180-km long coastline between Cape Lookout and Cape Fear, there are four rivers that discharge into the heads of estuaries (North, Newport, White Oak, and New), but > 60 tidal creeks positioned seaward of bayhead deltas, 8–30 km from the ocean shoreline. The small size of tidal-creek watersheds may be disproportionate to their importance as conduits for sediment transport from landscapes to estuaries and the coastal ocean because they are abundant, located in areas of expanding populations and creek discharge is strongly impacted by storminess. These abundant tidal creeks are the primary conduits of sediments and nutrients between lower coastal plain areas and coastal waters along the margins of estuaries and are commonly disregarded in estuarine sediment-flux models. While these coastal watersheds have little relief (<16 m), area (<32 km$^2$), and discharge, they are more impacted by population growth and associated land-use change, in terms of percent watershed area relative to watershed size. Unlike larger coastal plain and piedmont rivers, tidal creeks are not commonly dammed to form reservoirs that buffer the downstream estuary against land-use changes upstream. We mapped changes in land cover of tidal-creek watersheds, identified periods of greatest change, and compared those periods to the depositional record since 1900 CE using the radiometric tracer $^{210}$Pb. Linking changes in land cover to changes in sedimentation will provide guidance to managers (like those from the Erosion and Sediment Control Commission of NC) and stakeholders on how to increase the resilience of coastal communities through wetland and soil conservation practices, which could include dredging, installation or removal of buffers along creeks, or modifications to existing or future developments (e.g., reductions in impervious surfaces).

## Methods

### Site selection

The NC coast hosts extensive estuarine systems, a broad coastal plain, and recently modified coastal watersheds making it an ideal study area. The topography of the lower coastal plain is defined by a series of terraces and scarps emplaced during the Pleistocene glacial eustatic highstands [35]. The Suffolk paleoshoreline formed ~80 ka [36] at an elevation between 6 and 14 m, parallels the NC ocean shoreline from the southwest to Morehead City, where its trend abruptly changes to the North forming a right angle or paleo cape [36]. The paleo cape at Morehead City divides the study area into two distinct zones. North of the paleo cape, the coastal plain seaward of the Suffolk Shoreline is characterized by a low slope, elevation <2 m, and numerous tributary incised valleys [36]. The astronomical tidal range in estuaries decreases north of the paleo cape from 0.95 m at NOAA Station ID 8656483 near Morehead City in Back Sound to about 0.37 m near Cedar Island, NC in southern Pamlico sound (NOAA Station ID 8655151). Southwest of the paleo cape the Suffolk Shoreline is closer to the ocean creating relatively steep coastal plain slopes with Carolina Bays typically positioned landward of the Suffolk Shoreline, numerous coastal prism incised valleys, and a tidal range of 1.21 m at NOAA Station ID 8658163. Carolina Bays are oval shaped depressions (<3 m deep) with elevated rims that vary in size (~100m – 4 km along the long axis) along the southeastern Atlantic coast and are oriented NW to SE [37]. Our study includes 12 tidal creeks in southeastern NC, split between Carteret County (CC; lower tidal range) and New Hanover County (NHC; higher tidal range; Fig 1). Of those 12 tidal creeks seven are within coastal prism incised valleys, six in NHC and one in CC (Gales Creek, Site 6), and five are within tributary incised valleys, all in CC (Fig 1).

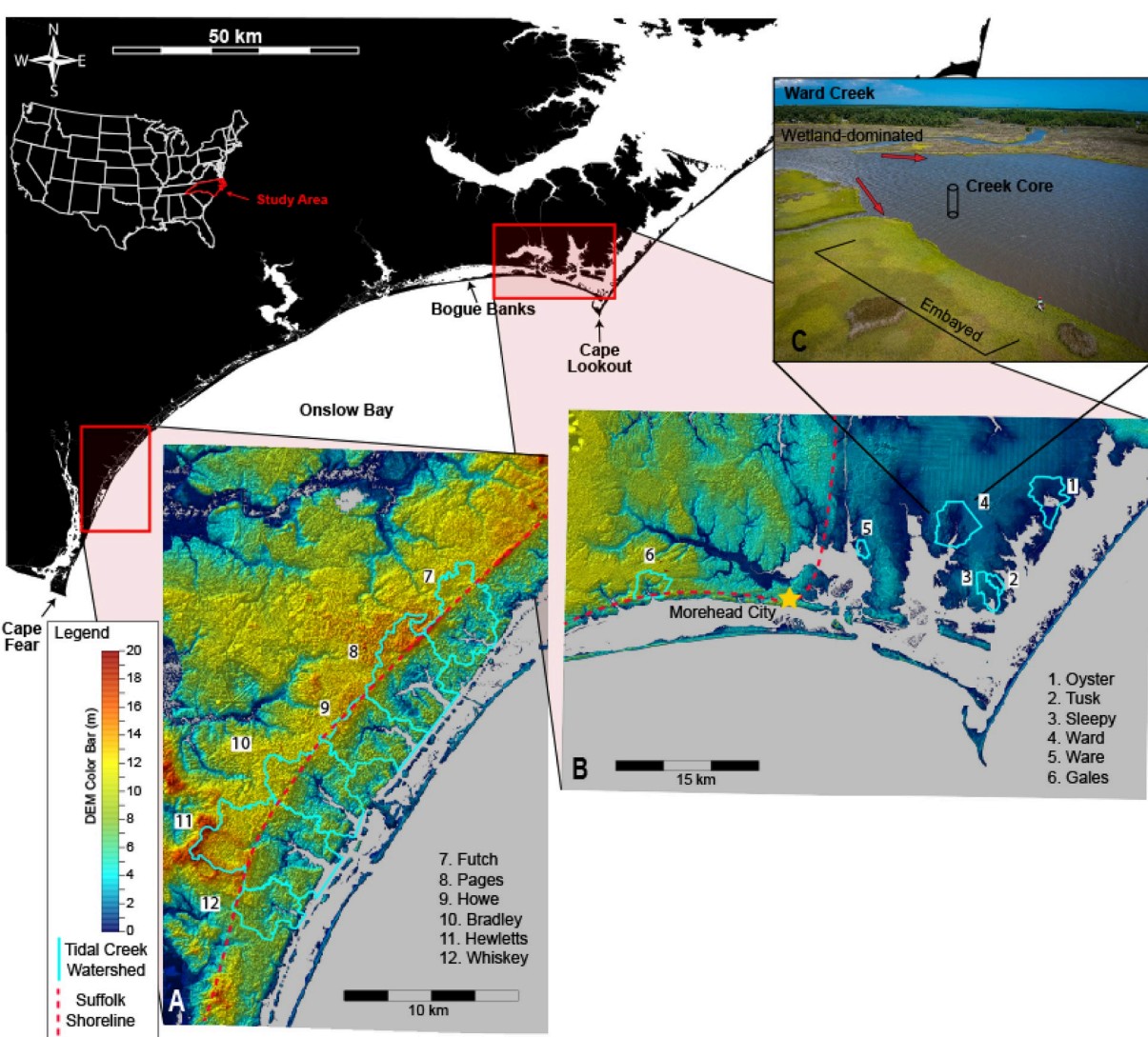

**Fig 1. Study area map.** The topography and watershed boundaries of each tidal creek with the creek site numbers increasing from east to west (1–12). Six creeks were sampled in both New Hanover County (A), and Carteret County (B). A core from each creek was obtained from the same general location, with respect to the creek outlet, as shown for Ward Creek (C). DEMs in A and B sourced from NOAAs North Carolina Sea Level Rise Digital Elevation Model (2006) with data <0 removed to include only topography without bathymetry data.

## Land-use change and digital elevation models

Watersheds were delineated by hand in ESRI ArcGIS using digital elevation models (DEMs) from lidar data collected in 2014 for CC (± 0.12 m vertical; [38]; Fig 1B), and NHC watersheds were obtained from the County Data Hub [39]. A DEM was obtained for NHC from lidar data collected in 2015 (± 0.12 m vertical; [40]; Fig 1A), and watershed relief in both counties was calculated as the difference between the highest and lowest 10% of elevation points of the DEM (Table 1, Fig 1A and 1B).

Based on imagery availability, land cover of each watershed area was classified at least every decade between the 1959–2016 study period (Table 2). From 1959–1993, land cover was digitized using georeferenced aerial imagery from the United States Geological Survey (USGS)

**Table 1. Land-cover datasets.** The years land-cover classes were mapped in each tidal creek watershed (Green) with missing land cover data in orange. See Fig 1 for creek numbering scheme.

| Year | 1 | 2 | 3 | 4 | 5 | 6 | 7 | 8 | 9 | 10 | 11 | 12 |
|------|---|---|---|---|---|---|---|---|---|----|----|----|
| 1959 | | | | | | | | | | | | |
| 1964 | | | | | | | | | | | | |
| 1969 | orange | orange | orange | orange | orange | orange | | | | | | |
| 1970 | | orange | orange | | | orange | orange | orange | orange | orange | orange | orange |
| 1975 | | | | | | | | | | | | |
| 1982 | | | | | | | | | | | | |
| 1993 | | | | | | | | | | | | |
| 1996 | | | | | | | | | | | | |
| 2001 | | | | | | | | | | | | |
| 2006 | | | | | | | | | | | | |
| 2010 | | | | | | | | | | | | |
| 2016 | | | | | | | | | | | | |

Aerial Photo Single Frames records collection and National High-Altitude Photography (NHAP) program. Classifications used were forest, cleared forest, agriculture, developed, or water/intertidal. Changes between cleared forest and forest classifications were associated with silviculture operations. Land-cover from 1996–2016 was obtained from the Coastal Change Analysis Program (C-CAP; [39]) and reclassified to match the same categories as the earlier

**Table 2. Watershed characteristics and core locations.** Creek name with corresponding site number, Dbs = Drainage basin size or watershed area, watershed relief, core collection date and location, and creek type (Tributary or Coastal Prism incised valley).

| Site ID | Creek Name | Dbs (km$^2$) | Watershed Relief (m) | Core Date | Core latitude (degrees) | Core longitude (degrees) | Approximate water depth (m) | Creek Valley Type |
|---------|-----------|--------------|----------------------|-----------|-------------------------|--------------------------|------------------------------|-------------------|
| 1 | Oyster | 11.76 | 2.30 | 2016-08-05 | 34.8270 | -76.4623 | 1.1 | Tributary |
| 2 | Tusk | 1.88 | 2.68 | 2016-08-09 | 34.7463 | -76.5139 | 1.2 | Tributary |
| 3 | Sleepy | 5.38 | 3.73 | 2016-07-21 | 34.7329 | -76.5272 | 0.8 | Tributary |
| 4 | Ward | 14.96 | 2.44 | 2016-07-07 | 34.7907 | -76.5658 | 1.1 | Tributary |
| 5 | Ware | 1.54 | 2.49 | 2016-08-24 | 34.7750 | -76.6741 | 0.8 | Tributary |
| 6 | Gales | 7.78 | 9.48 | 2016-07-28 | 34.7321 | -76.9075 | 1.2 | Coastal Prism |
| 7 | Futch | 15.44 | 11.88 | 2016-05-10 | 34.3041 | -77.7532 | 0.75 | Coastal Prism |
| 8 | Pages | 20.35 | 14.60 | 2016-10-04 | 34.2816 | -77.7786 | 1.1 | Coastal Prism |
| 9 | Howe | 14.24 | 13.47 | 2016-10-05 | 34.2559 | -77.8065 | 1.1 | Coastal Prism |
| 10 | Bradley | 18.67 | 12.57 | 2016-09-20 | 34.2179 | -77.8395 | 1 | Coastal Prism |
| 11 | Hewletts | 30.23 | 15.08 | 2016-09-12 | 34.8270 | -76.4623 | 1.5 | Coastal Prism |
| 12 | Whiskey | 8.49 | 8.74 | 2016-09-12 | 34.1600 | -77.8626 | 1.1 | Coastal Prism |

time-steps. Some of the aerial imagery data sets did not capture every watershed, which resulted in land cover being classified 11 times for nine tidal creeks and 10 times for three tidal creeks (Table 2).

The percent watershed area of each land cover class changes over time; however, there were typically two classes that changed the most over the 50-year period, with one class being replaced by the other. We defined the major land-cover change (MLCC) for a watershed as the class with the highest percent contribution to the total change in land cover over the 50-year period. For every watershed, we defined the time boundary between pre- and post-MLCC as the midpoint between the dates of two land cover classification maps where a $\geq$15% change occurred within the record over ~10-year period (S1 Fig). The 15% change criterion is the 90th percentile of all the changes recorded between successive land cover classification maps for the MLCCs.

## Field sampling and radiometric dating

To maximize the potential for obtaining a complete high-resolution sedimentary record, each tidal creek was sampled where the channel widened and transitioned from wetland-dominated to embayed or within the bayhead shoreline ([41]; Fig 1C). A push core (1 m) using an aluminum tube of 10.16 cm diameter, ~1 m long, and wall thickness of 1.3 mm was collected in 2016 from each creek bottom (water depth ~1m; Table 2) with the aid of a sledge-hammer. Sediment cores were transported back to the laboratory in a vertical position, extruded into 1-cm intervals (~10 cm$^3$) and frozen. Samples were then weighed, freeze-dried, and weighed again to calculate porosity. Dry bulk density values were calculated assuming quartz composition. Disaggregated subsamples were used to measure percent organic matter by loss on ignition (LOI; [42]) and grain size (<2000 to 0.04 μm fraction) using a Cilas laser particle size analyzer. The remaining samples were then sorted through a 63-micrometer sieve for radioisotope analysis.

The fine-grained fraction (<63 microns) was used for isotope-dilution alpha spectrometry to quantify $^{210}$Pb via the granddaughter isotope, $^{210}$Po, which is assumed to be in secular equi-librium with $^{210}$Pb [43–46]. Raw $^{210}$Pb data and dry bulk density for each depth interval for all 12 creek cores, both of which are needed for geochronology modelling, are presented in the attached S1 Data. To obtain accumulation rates, $^{210}$Pb dating methods including the Constant Flux and Constant Flux Constant Sedimentation models described in Sanchez-Cabeza and Ruiz-Fernandez (2012) were used. For this study, the Constant Flux model (CF; widely known as the Constant Rate of Supply model; [47, 48]) was of the most interest because it assumes that $^{210}$Pb flux to the sediment surface is constant and the initial concentrations and the mass accumulation rates (MAR) of individual layers may change, but they must be inversely propor-tional. The CF model, therefore, is useful for tracking sedimentation rates that vary over time. Depth integrated $^{210}$Pb inventories were calculated for each sediment core and used to calcu-late sediment accumulation rates (SAR) with the CF model. This model provides ages and accumulation rates for discrete intervals within the core allowing for direct comparison with the land-cover time-series data. All reported p-values are the result of t-tests assuming unequal variances between sample groupings.

To determine whether an acceleration in MAR was associated with the period between pre and post MLCC, we needed to identify the initial time a major acceleration in MAR occurred. Starting from 1959, the beginning of the land cover dataset and moving towards present, we calculated the per-cent change in MAR between each measurement and defined the initial major acceleration in MAR as >10% increase or when MAR switched from decelerating to accelerating.

## Results

### Drainage basin size and relief

The drainage basins of the tidal creeks vary in area and relief by an order of magnitude. Coastal prism drainage basins are generally larger and have greater relief than the tidal creeks within tributary incised valleys (Table 1) with Hewletts Creek (Site 11) having the largest area (30.23 km$^2$) and relief (15.08 m), Ware Creek (Site 5) having the smallest area (1.54 km$^2$), and Oyster Creek (Site 1) having the lowest relief (2.3 m). Drainage basin size increases with relief and is a power function of relief excluding two outliers, Oyster (Site 1) and Ward (Site 4) creeks (Fig 2). Both outliers are tributary incised valleys with larger drainage basins than what would be predicted by their low reliefs. Ward Creek is positioned along the flank of the North River incised valley, which is 8–10 m deep at the confluence [26]. When sea level was lower and the area was exposed, the relic steep slope and high relief along the flank of the North River incised valley promoted incision, knickpoint migration, and expansion of the Ward Creek drainage basin above what would be predicted from watershed relief [26]. The other tidal creeks that formed within tributary incised valleys merge with their associated larger river systems on the inner continental shelf, farther seaward than where Ward Creek merges with the North River [49], and the size of those drainage basins, at the present headwater locations, is controlled mainly by relief. The Oyster Creek outlier has the lowest watershed relief and a disproportionately large drainage basin area because of human modifications. Extensive ditching of nearby farmland and wetlands and creation of a 1.25 km$^2$ waterfowl impoundment site increased watershed area above its natural state.

### Land cover 1959 to 2016

Land cover of all 12 tidal creek watersheds was dominated by forest, cleared forest, and agriculture in the 1959 imagery (Fig 3A). During the next 57 years, development of the creek watersheds in NHC increased to an average of 50.2 ±5.5% of total watershed area, while the CC creek watersheds remained dominated by forest, cleared forest, and agriculture (Fig 3B). The creek watersheds in NHC are adjacent to the expanding Wilmington, NC urban center and developed into suburban landscapes, while the creek watersheds in CC remained relatively rural over the same period. Development of the NHC creek watersheds, the MLCC class, increased the most, ~20–26%, between the early 1970s and middle 1990s (S1 Fig). Initiation of MLCC generally occurred first in the watersheds closer to Wilmington, NC; ~1964 for Whiskey Creek in the west and ~1989 for Futch Creek in the east. In the tidal creek watersheds of CC, the MLCC class was cleared forest except for Ward Creek (Site 4) where the MLCC class was agriculture (S1 Fig). The initiation of MLCC (cleared forest) of the Oyster Creek watershed (Site 1) occurred between 1982 and 1993. The percent area of forest and cleared forest fluctuated after 1975 in the Tusk Creek (Site 2) and Sleepy Creek (Site 3) watersheds, mainly the result of silviculture operations, with the start of MLCC occurring between 1975 and 1982 with 18% and 28% increase in cleared forest at each respective site. Agriculture area rapidly increased between 1975 and 1982 in the Ward Creek watershed (Site 4). Land cover of the Ware Creek watershed (Site 5) changed the least throughout the period in all classes and never exceeded the 15% threshold for the initiation of MLCC, with a maximum increase of only 11% cleared forest between 1993 and 1996. The land cover of the Gales Creek watershed (Site 6) increased 15% in developed area between 1964 and 1982, the highest increase in development of all the CC sites, and forest and cleared forest area fluctuated after 1964. The MLCC class in Gales Creek, however, was cleared forest which decreased 38.4% in area between 1964 and 1975 and was associated with a 35% gain in forest area (S1 Fig). Increasing forested area is an

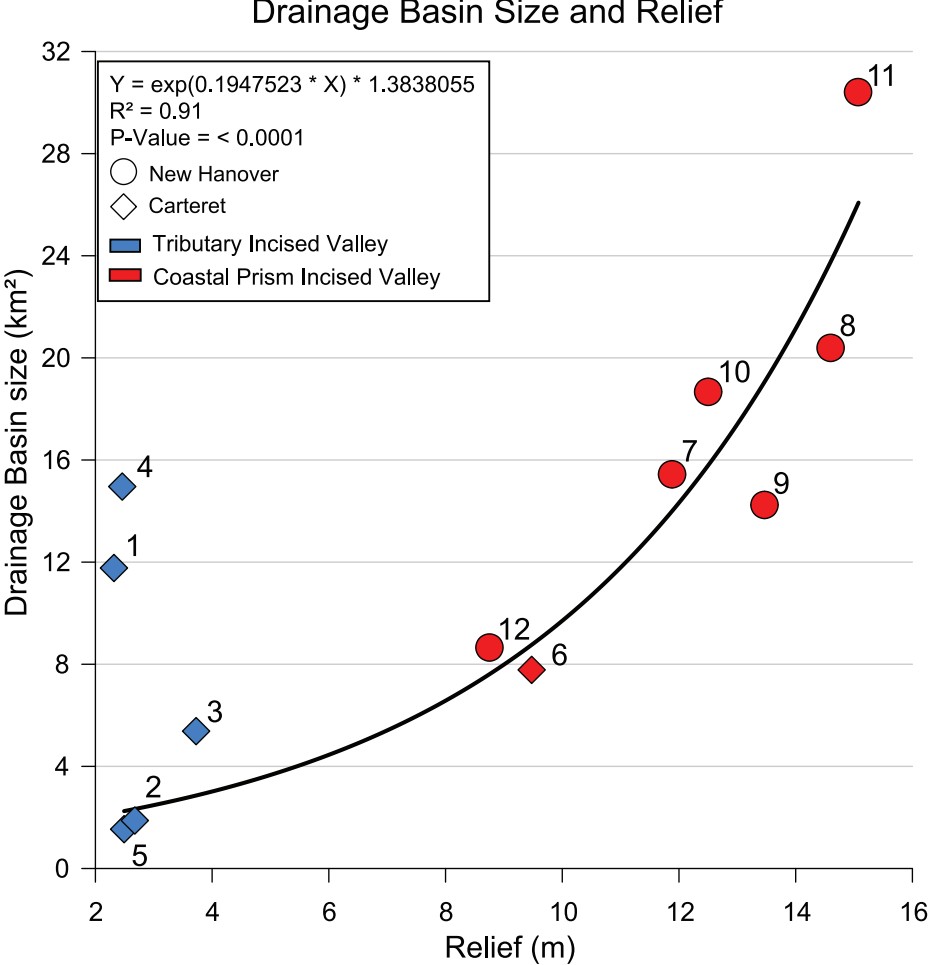

**Fig 2. Drainage basin size vs relief.** Tidal creek drainage basin size increases exponentially with relief. Tidal creeks are enumerated as shown on Fig 1. Oyster Creek (1) and Ward Creek (4) are outliers excluded from the regression.

unlikely mechanism for mobilizing sediment. Despite the decrease in cleared forest area fitting our criteria for the MLCC class, the smaller contemporaneous increase in development is more relevant for impacting sedimentation downstream.

### Tidal creek sediment composition

Changes in sediment composition and texture with depth in a sediment core can indicate changes in sediment source or support a hypothesized change in sedimentary regime. Most of the tidal creek cores showed minor variation in sediment composition with depth, but there were some differences in texture between CC and NHC. The CC creek sites generally had less sand than the NHC creek sites (Fig 4). The broad differences in sediment texture between counties can be explained by the distinct depositional environments over which the watersheds formed, capacity for transporting sand, and sediment texture of adjacent estuarine source areas. The tidal creek watersheds in NHC are positioned on the sandy relatively high-relief Suffolk Scarp paleoshoreline, while most of the tidal creek watersheds in CC formed on old inner-continental shelf muddy substrate with low relief [49]. The Gales Creek watershed in CC (Site 6) drains a low relief area of the Suffolk Scarp and the core lacks sand. Importantly,

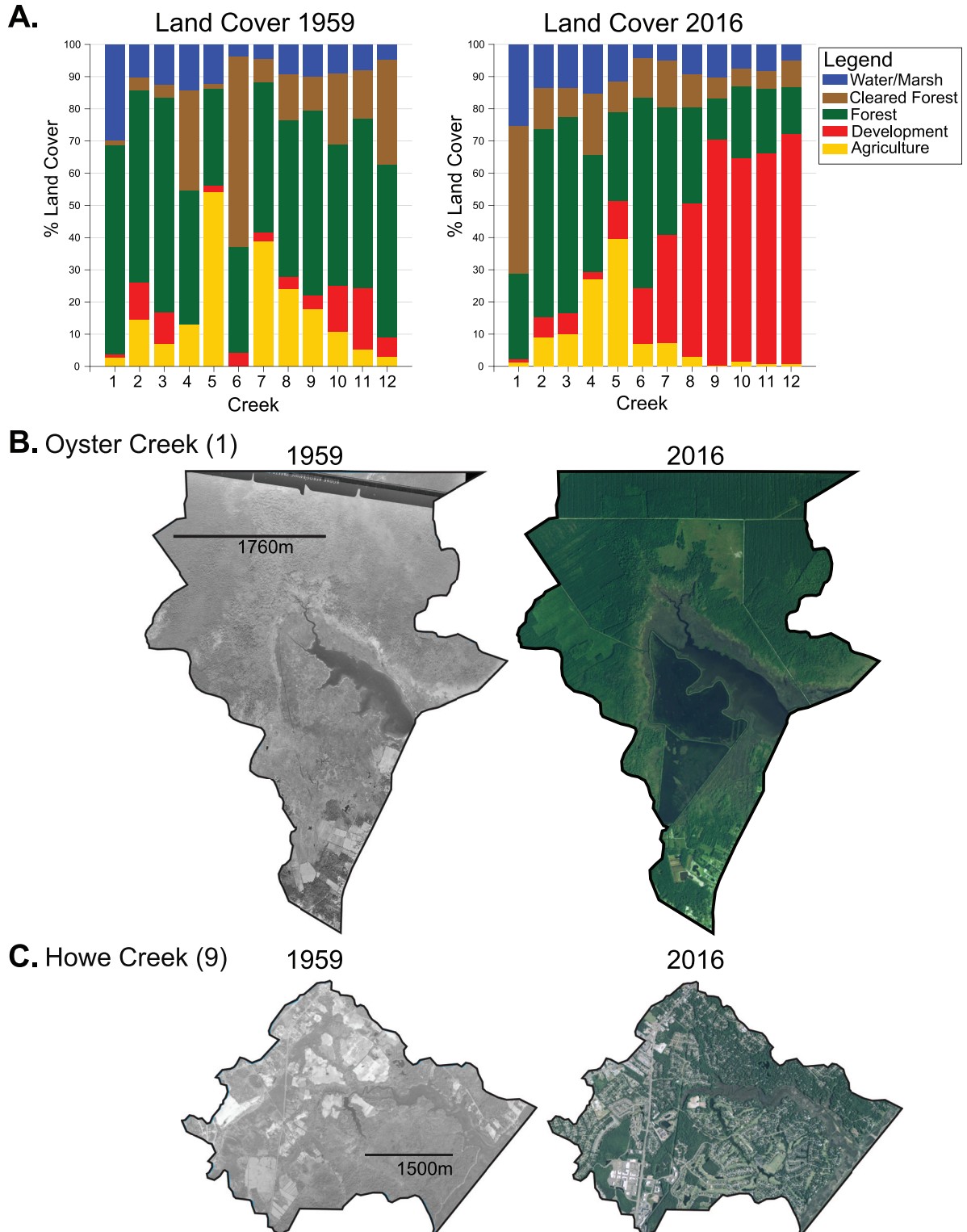

**Fig 3. Earliest land-cover and most recent land-cover.** Tidal-creek watershed percent land cover of each class in 1959 and 2016 (A). Agriculture was the MLCC for Oyster Creek (B) and development was the MLCC for Howe Creek (C).

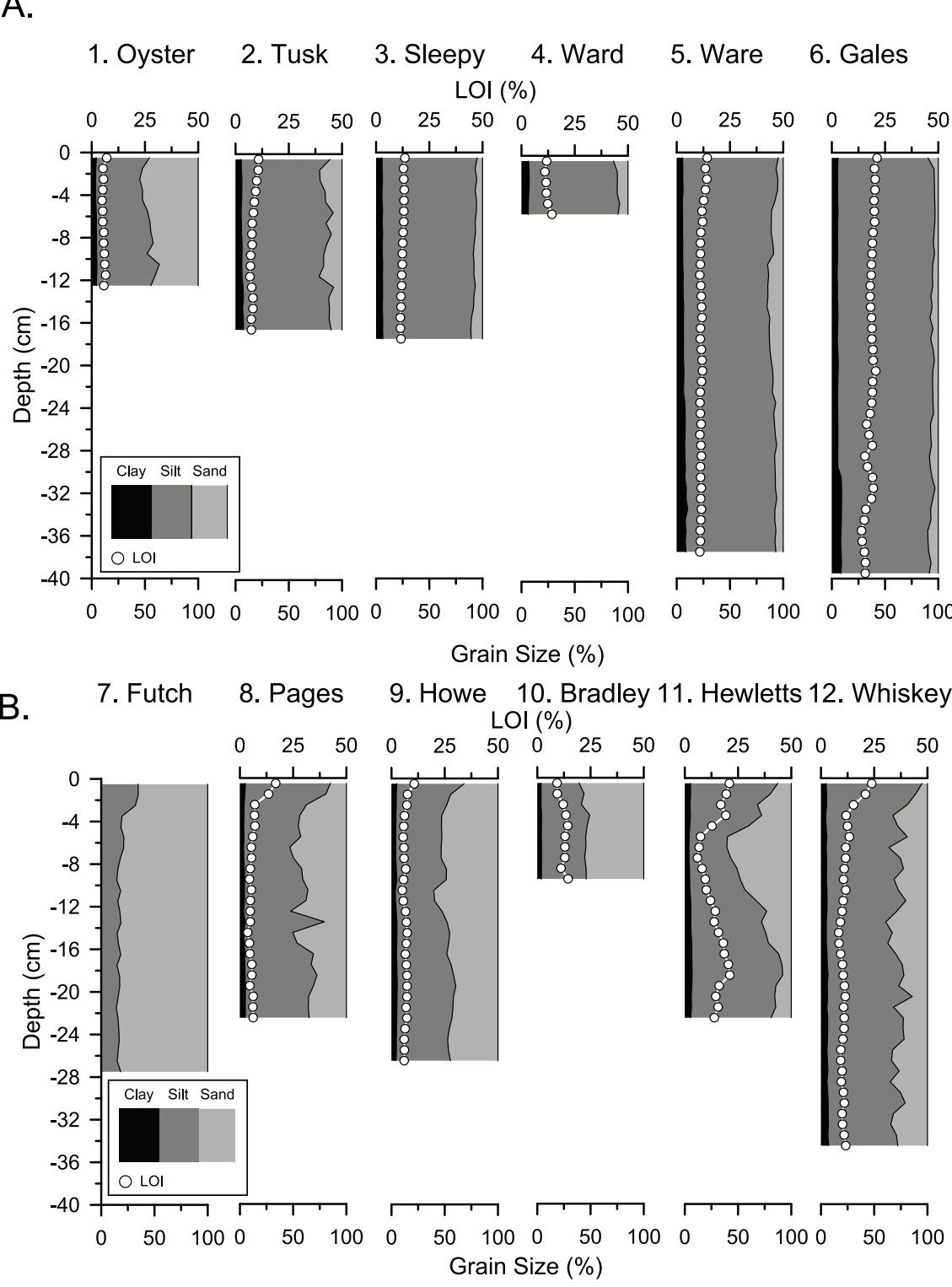

**Fig 4. Changes in the composition of each core with depth.** CC creek sites in A and NHC creek sites in B. Cumulative percent of clay (black), silt (dark grey), and sand (light grey) and percent organic matter (while circles) are shown for each 1-cm thick core sample. Mid-point depths are plotted from the top of the bed (0 cm) to the base of the $^{210}$Pb profile.

the embayments of the NHC creeks are positioned closer to modern sandy tidal deltas and have a greater tidal range and capacity for transporting sand than CC creeks. The one exception is Oyster Creek (Site 1) that had between 37–55% sand (Fig 4) throughout the sediment core. Of the CC creek sites, Oyster creek has the most direct connection to sandy Core Sound [50]. There was also relatively little change in percent organic matter with depth for most of the cores (Fig 4). The top ~3 cm of the cores from the NHC sites had a higher % organic matter than what was measured below, likely due to incomplete biodegradation of surface organic carbon. Hewletts Creek (Site 11) showed some variation in organic matter and sediment composition downcore, which could be attributed to a change in sediment regime (Fig 4).

## Mass accumulation rates through time

Long term average (1900–2016) MARs were higher in tidal creeks within coastal prism incised valleys than tributary incised valleys (0.21 ± 0.03 g cm$^2$ yr$^{-1}$ and 0.12 ± 0.03 g cm$^2$ yr$^{-1}$, respectively, P = 0.033) excluding Bradley Creek (Site 10) which was mixed throughout the profile making accumulation rates impossible to derive. Average MAR from 1900–1950 was the same in both creek valley types, but from 1950–2016, average MAR was higher in creeks within coastal prism incised valleys than tributary incised valleys (0.22 ± 0.004 and 0.14 ± 0.002 g cm$^2$ yr$^{-1}$, respectively, P = 0.032). Most creeks showed an increasing trend in MARs from 1900 to 2016 and most of the increase occurred in the second half of the century (1950–2016), the only exception being Ward Creek (Site 4) with consistently low MARs (Fig 5A).

The MARs of Oyster, Tusk, Sleepy, Gales, Futch, Pages, Hewett's, and Whiskey creeks (Sites 1–3, 6, 7, 8, 11, and 12) rapidly increased following the mid-1970s or early 1980s, reached a maximum in the 1990s or early 2000s and subsequently stabilized or decreased. In contrast, the MARs at Howe (Site 9) and Ware (Site 5) creeks increased until ~1960, the beginning of our land-cover time-series data set, but then showed small fluctuations until 2016.

## Connectivity between watershed MLCC and MAR

The MARs of 8 creek sites increased during the period of the land cover record (1959–2016). One of the 12 sites in this study (Bradley, Site 10) did not yield a usable sediment record due to mixing and another creek watershed (Ware, Site 5) did not experience a MLCC as we have defined it in this study. Therefore, only 10 watersheds and associated core locations could be used to directly compare MARs to MLCC.

Relating changes in MAR with changes in land cover using more rigorous quantitative methods is problematic because it would require us to coarsen the temporal resolution of the accumulation rates to match the decadal land cover record, removing clear trends in accumulation rates that exist at the sub-decadal time scale (Fig 6). However, the timing of the increase in MAR after 1959 (>10% change between two periods) corresponded with the initiation of MLCC at eight of the creek sites (Sites 1–3, 6–8, 11, and 12), cleared forest for the creek sites within tributary incised valleys and suburban development for creek sites within coastal prism incised valleys (Fig 6). Howe Creek (Site 9) and Ward Creek (Site 4) showed no increase in MARs >10% since 1959, despite a watershed MLCC. Ware Creek (Site 5) experienced little change in land cover since 1959, always below the 15% threshold defining the initiation of a MLCC, but preserved some variability in sediment accumulation rates, most within error, which aligned with the timing of small changes in land-cover (Fig 6).

Sites 1–3 experienced variations in cleared-forest area from silviculture activities that started in the mid-1970s through the mid-1980s. Immediately after the initiation of silviculture, MAR accelerated. The creek sites impacted primarily by changes in silviculture recorded a more persistent acceleration in MAR as forested and cleared forest areas transitioned back

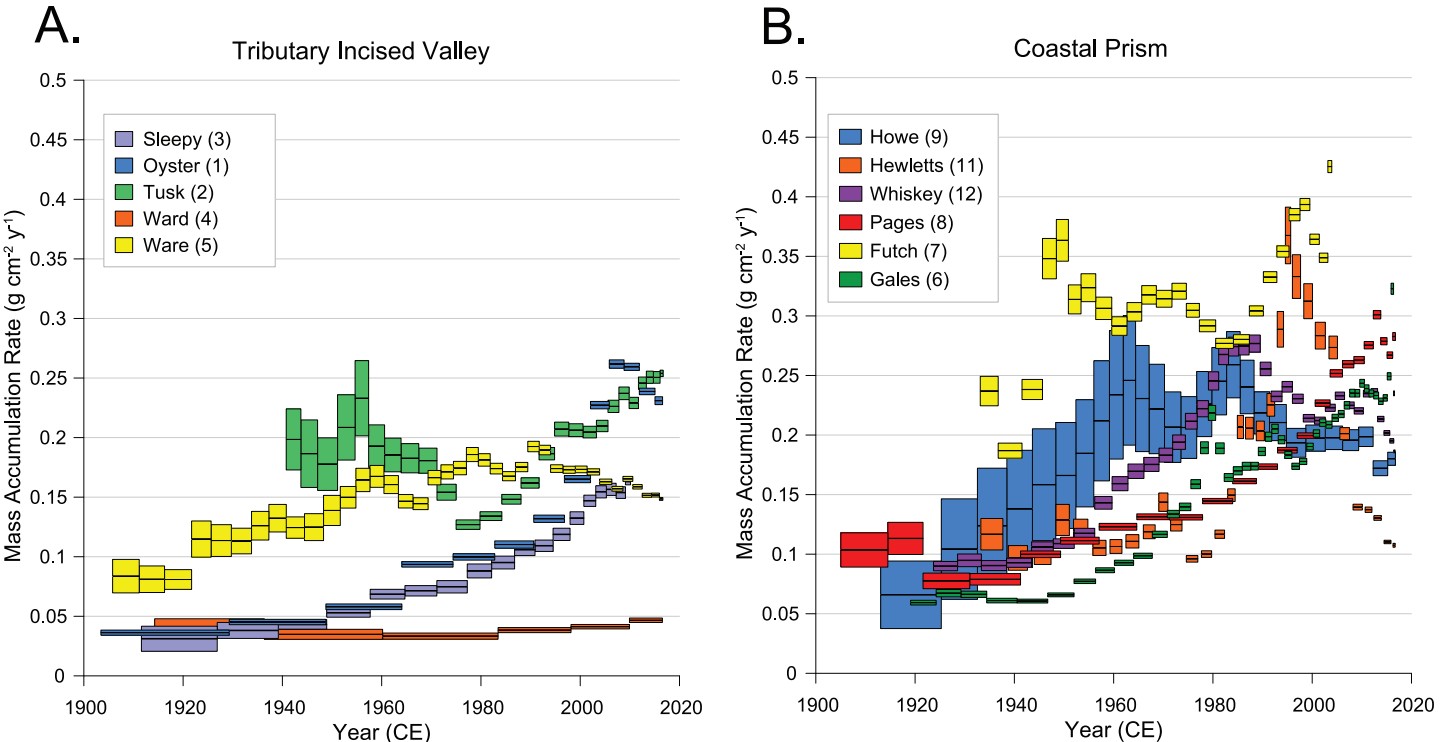

**Fig 5. MAR since 1900.** Mass accumulation rates (MAR) for cores from tidal creeks within tributary incised valleys (A) and coastal prism incised valleys (B). Bradley Creek could not be analyzed due to mixing. Vertical errors are measurement uncertainty representative of alpha spectrometry and horizontal error shows the amount of time each 1-cm slice represents in the core.

and forth with reestablishment and harvesting of pine trees. The land cover of the Gales Creek watershed (Site 6) experienced the largest change between 1964 and 1975 with a decrease in cleared forest area and an increase in developed area (Fig 6). These land-cover changes corresponded with accelerating MARs between 1964 and 1975 and a transition from silviculture to development and agriculture in the watershed. The MARs of sites 7, 8, and 12 (Futch, Pages, and Whiskey creeks, respectively) accelerated rapidly during periods of increasing development in the late-1980s, mid-1970s, and mid-1960s, respectively.

### Average SAR pre and post MLCC

Sediment accumulation rates (SAR) provide an indicator of changes in bed elevation, and in coastal depocenters SAR typically match rates of relative sea-level rise over long timescales [42]. Half of the creek sites experienced a doubling or more in average SAR post-MLCC, but all sites except Howe Creek (Site 9) showed an acceleration in SAR post-MLCC (Fig 7), despite the difference in type and percent land cover change between NHC and CC creeks. This widespread increase in average SARs can be attributed to increases in sediment load and increased accommodation through RSLR. There is a gradient in average 20th century rates of RSLR along the NC coast, increasing toward the north from $2.1 \pm 0.5$ mm yr$^{-1}$ at Wilmington, near the NHC creeks to $3.5 \pm 0.3$ mm yr$^{-1}$ at Tump Point, near CC creek sites [51]. Most sites recorded average SARs >3.5 mm yr$^{-1}$ and up to ~8.4 mm yr$^{-1}$ (Gales Creek, Site 6; Fig 7) post-MLCC, suggesting that these tidal creeks are infilling and getting shallower. Accommodation limits SAR and some sediment is likely bypassing and entering the adjacent estuary in CC and

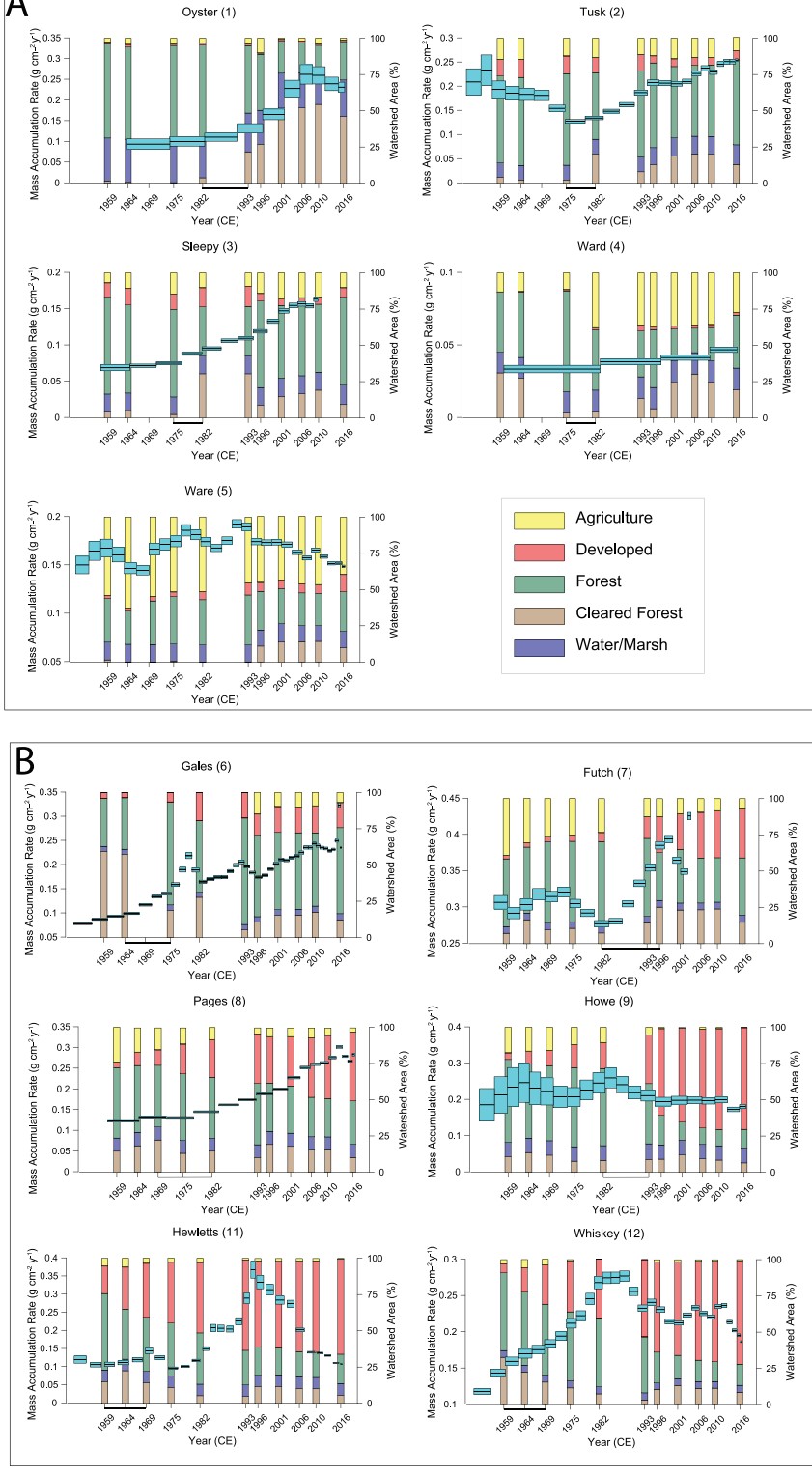

**Fig 6. MAR since 1950 vs land-cover change.** MAR (cyan) and tidal creek watershed land cover since 1950 as seen in S1 Fig. Tributary incised valleys (A) and Coastal Prism incised valleys (B) with MLCC as bold bar between the two land-cover time steps with >15% change. The earlier part of the sedimentary record is shown in Fig 5.

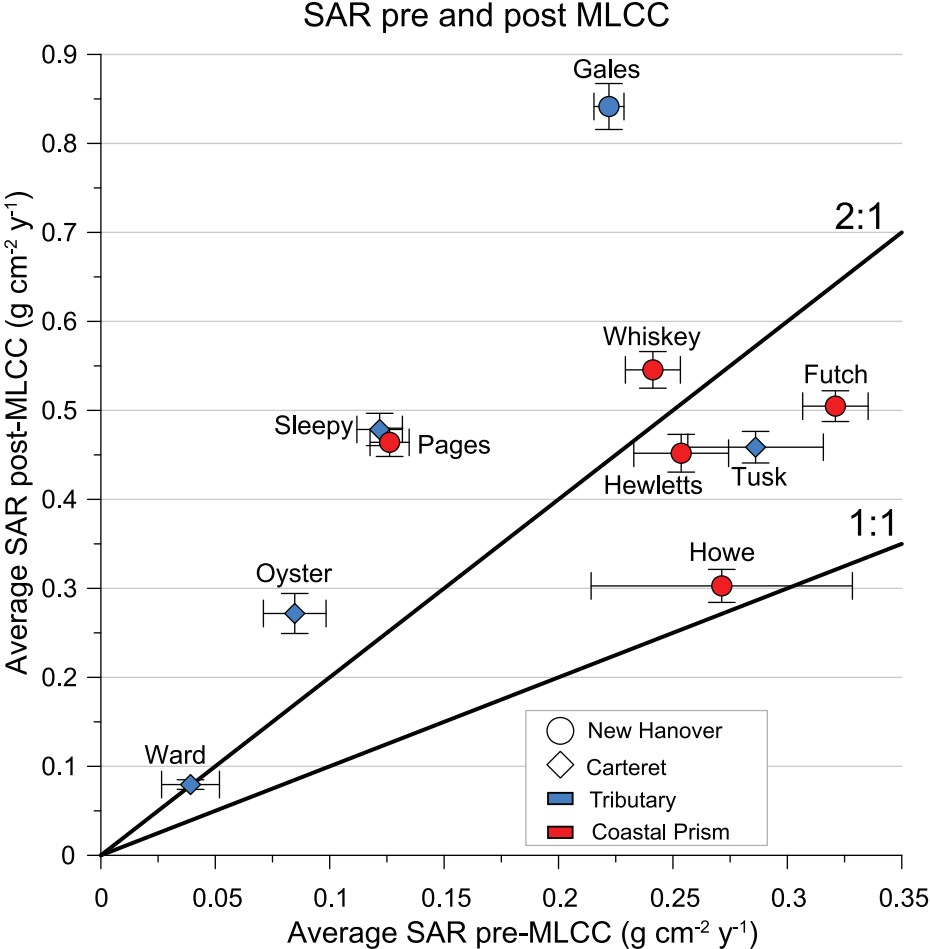

**Fig 7. SARS Pre vs Post MLCC.** Average Pre and Post MLCC sediment accumulation rates compared to a 2:1 and a 1:1 line. Error bars represent average measurement error.

the coastal ocean in NHC, especially during storm events, which could be contributing to higher MAR and SAR seen regionally in larger estuarine depocenters [22].

## Discussion

The MARs of the tidal creeks were higher at sites within coastal prism incised valleys than tributary incised valleys. The difference in MARs between valley types could, in part, be a result of accommodation differences between coastal prism and tributary incised valleys. Tributary incised valley systems have an oversized embayed area for sediments to accumulate in. Any commensurate increase in sediment flux would result in deposition of a thinner layer and lower recorded accumulation rates in tributary incised valleys than coastal prism incised valleys. In addition, tidal creeks within coastal prism incised valleys have a greater percent of intertidal area and associated oyster reefs than tidal creeks within tributary incised valleys, and oysters promote deposition through filter feeding and the production of feces and pseudofeces.

Constraining the relative proportions of the various sediment sources that contributed to the increase at each creek is not possible with our data set; however, given that we included

numerous tidal creeks in the study, many of which are in proximity to each other, it is possible to identify likely forcing mechanisms. The cores collected for this study were all from tidal creek outlets or bayhead shorelines [41] where an increase in MAR eventually forms tidal flats that become colonized with salt marsh as the creek outlet moves basin-ward. At the bayhead shoreline of tidal creeks, an increase in accumulation rate of the bed must be associated with an increase in sediment load and/or a decrease in the depth of bed erosion. Tidal creeks receive some sediment from larger rivers through their connection with estuaries and the coastal ocean and from shoreline erosion. If an increase in sediment supply to those connected water bodies caused the increase in MAR measured at most of our sites, then the timing of the increase should correspond with the sediment loading of those larger rivers and the increase recorded as a synchronous shift in depositional regime of many adjacent tidal creeks. There is no hydrological connectivity between the NHC tidal creek sites and the nearby Cape Fear River, as they do not share a watershed, but sediment loading in the Neuse and the Newport rivers could have influenced sedimentation in the CC tidal creeks. The timing of the increase in MAR for the tidal creeks in CC, however, occurred after 1975 CE, postdating the increase in sediment loading of the Newport River (1964; [3]) and the Neuse River (~1940; [22, 52]; Fig 6).

Coastal rain events and flooding from storm surge disproportionately affect tidal creeks as compared to large river systems due to their low elevation, small relief, and position along the shoreline. Changes in coastal storminess can promote shoreline erosion and increase sediment supply; however, it would also increase the depth of erosion at the creek outlet and increase subtidal area. Furthermore, the proximity of the sites to each other suggests that a change in storminess would also result in a simultaneous change in MAR, but the timing of the increase differs among creek sites. An increase in storms with wind $> 55.5$ km h$^{-1}$ (primarily northeasters) did occur around 2000 in the area near the CC sites [53] but increased storminess postdates the diachronous increase in MAR among creeks (Fig 5). Transport of sediment by sheet flow in tidal creek watersheds occurs during storms and coastal NC has entered a wetter climatic regime where storm events are bringing more extreme precipitation patterns [54, 55]. The predicted increase in storm intensity and frequency in the region [56], should result in further increased connectivity between tidal creek watersheds and downstream depositional environments.

The timing of the increase in MARs after 1959 corresponded with the initiation of the MLCC period at all of the creek sites where there was both a usable sedimentary record and a MLCC. While all the core sites recorded an increase in MAR commensurate with the initiation of a MLCC, many of the sedimentary records showed subsequent maintenance of constant or decelerating MARs despite there being no further large changes in land cover of the creek watersheds. It is likely that surface sediment was eroded from the watershed and transported to the creeks during the construction phase of development, which included clearcutting forest, and once developed there was little sediment available that could be mobilized and exported to the tidal creeks.

Hewletts Creek (Site 11), recorded an increase in MAR >10% at the time of MLCC; however, a greater increase in MAR occurred after that period. Development of the Hewletts Creek watershed initiated in the 1950s, but the increase in MAR was not recorded until ~1980. The pattern of development across the Hewletts Creek watershed progressed from landward to seaward, initiating in the upper part of the watershed within a Carolina Bay depression (~3.5 km$^2$ within a 30.2 km$^2$ watershed; S2 Fig). This depression, with a central man-made retention pond, likely acted as a depocenter for ~11% of the total Hewletts Creek watershed, limiting drainage to the lower watershed and creek. The increase in development prior to 1982 occurred mostly within the confines of the Carolina Bay depression and was not recorded as

an increase in MAR because the depression disconnected the modified landscape from the creek. The increase in development between 1982 and 1993 occurred outside of the Carolina Bay depression throughout the lower watershed and MAR accelerated rapidly during the same period. The core from Hewletts Creek also showed a decrease in organic matter and an increase in sand between 1982 and 1996. Only Hewletts Creek recorded changes in sediment composition as MAR accelerated and unlike the other sites, most of the increase in development occurred on the sandy Suffolk Scarp, which makes up 46% of the 30.23 km$^2$ watershed.

The land cover record of Ward and Howe creeks (sites 4 and 9, respectively) contained a MLCC, but no commensurate change in MAR was observed. The NE trend of Ward Creek (Site 4) makes it susceptible to resuspension and flushing during SW wind events, the prevailing wind direction in the area. The Ward Creek sedimentary record consistently recorded the lowest MARs of all sites and unsupported $^{210}$Pb was only measured down to 7 cm in the core, indicating the orientation of the creek is conducive to sediment bypass near the bay head shoreline, not deposition. Large changes in land cover of the Ward Creek watershed post MLCC in 1982, such as the increase in cleared forest from 6–24% from 1996–2001, likely did promote an increase in sediment flux to coastal areas; however, the core was not obtained from an efficient depocenter where that would be recorded. The Howe Creek (Site 9) sedimentary record also preserved little changes in MAR since 1959; however, the development area of the watershed more than doubled between 1982 and 1996. Unlike the other creeks in NHC that showed an increase in MAR as developed area of the watershed increased, the Howe Creek watershed had a relatively large area dedicated to retention ponds, >2% of the watershed as compared to <0.5% in other creeks. The retention ponds functioned to interrupt runoff from reaching downstream areas and likely restricted sediment transport following land-cover changes in the Howe Creek watershed.

The two tidal creeks with high-resolution sedimentation records that did not record an increase in MAR near the bay head shoreline (Sites 5 and 9), despite a watershed land-cover change, provide guidance on possible strategies for mitigating sediment loading. Land cover change of the Ware Creek watershed impacted <11% of the area and no corresponding increase in MAR was recorded indicating that small-scale land-use changes likely affect sediment loading less than large-scale changes. A natural basin functioning as a large retention pond and constructed retention ponds in association with large-scale development activities, mitigated excessive sediment loading to downstream habitats at Hewletts and Howe creeks, respectively. Retention ponds likely interrupted sediment transport pathways to downstream areas in both Hewletts and Howe creeks and should be applied more widely in development plans for coastal communities. However, there may be a threshold where too much sediment stored in watershed retention ponds limits supply to downstream intertidal habitats like salt marshes and oyster reefs and their ability to persist with RSLR.

## Connectivity between creek watersheds and downstream estuaries

Connectivity between tidal creek watersheds and downstream areas is dependent on sediment transport capacity. The stream power of tidal creeks is low because of their low slope, low discharge, and large backwater effects from lunar tides [57]. Despite a low sediment transport capacity, the connectivity of tidal creek watersheds to downstream bay head shorelines in this study was high, using the time lag between when erosion of the landscape from when a MLCC initiated and when that sediment began to accumulate at the bay head shoreline as a proxy [3]. The timing of the MLCC was contemporaneous with increases in MAR near the bayhead shoreline at eight of the tidal creeks, within the resolution of the remote-sensing data set (5–10 years). Land-use change is known to promote increases in sediment flux in low-relief areas

[3, 29], but connectivity and the magnitude of the increase tends to be greater for smaller systems with higher relief [18, 58, 59]. Creek watersheds within coastal prism incised valleys should be more connected to downstream bayhead shorelines than tributary incised valleys given their generally larger drainage basins and higher relief; however, this was not resolved with our data set. Oyster Creek (Site 1) is within a tributary incised valley, has the lowest relief watershed (2.3 m), and like the creeks within coastal prism incised valleys, the increase in MAR at the bayhead shoreline of Oyster Creek corresponded immediately with the increase in cleared forest area from silviculture activity (Fig 6). The reach of the tidal creeks from headwater to mouth is short, only 5–15 km for the creeks in this study, and the proportion of the watersheds that experienced a change in land cover was high (>50%) for most of the creeks. The short length of the creeks and large percent change in land cover explains why these small watersheds are highly connected to downstream bay head shorelines and we see no correlations between connectivity and drainage basin size or relief. Whiskey Creek (Site 12) has the smallest drainage basin area and relief in NHC; however, it recorded similar trends in MAR as adjacent Hewletts Creek (Site 11), both accelerating with an increase in development of the watersheds, despite Hewletts Creek having >three times greater drainage basin area and relief. In addition, the ~30% greater tidal range in NHC, as compared to CC, had little influence on connectivity. The MARs of Gales Creek, the only CC creek within a coastal prism incised valley accelerated with an increase in development after the MLCC, like the creeks in NHC (Fig 6).

## Conclusion

Despite variations in setting, watershed area and relief, and tidal range, tidal creek watersheds are directly connected to downstream bay head shorelines. Our data set cannot be used to explicitly identify sediment sources; however, MARs measured downstream, near the bay head shoreline, accelerated immediately (within our temporal resolution) after large changes in land cover occurred in the watershed at eight sites. This indicates sediment transport pathways in tidal creeks are more direct and responsive to land cover change than what their low relief and small watersheds would suggest. We documented a 10-15-fold increase in developed watershed area at four of the sites in NHC. Prior to MLCC, the average MAR at the sites were similarly low. At the end of the 20th century, the sites within coastal prism incised valleys, where development was the MLCC, showed higher average MARs than lower relief smaller tidal creeks within tributary incised valleys, where silviculture was the MLCC. The type of MLCC and/or watershed morphology influence the amount of sediment accumulating in down-stream coastal areas, and watersheds with small-scale changes in land cover or large changes in land cover with retention ponds showed little changes MAR at the tidal creek outlet. Most of the tidal creek bay head shoreline areas are accumulating sediment faster than the rate of RSLR; however, as RSLR continues to accelerate and these watersheds have little additional space to accommodate future changes in land cover, this is likely a temporary state. Tidal creeks discharge directly into the flood plain of larger river systems, estuaries, or the coastal ocean and their contribution to coastal sediment budgets is largely ignored but could partly explain the increase that occurred after 1950 at many sites across North America.

## Supporting information

**S1 Fig. Land-cover datasets since 1959.** Land cover classes shown as percent area of tidal creek watershed for each site in Carteret County (A) and New Hanover County (B). Land cover was classified at least once per decade. Bars connecting land cover datasets indicate

MLCC for each creek watershed.
(EPS)

**S2 Fig. Carolina Bay within Site 11.** Digital elevation model of Hewletts Creek (Site 11) highlighting the Carolina Bay with a central retention pond in the northwestern part of the watershed.
(TIF)

**S1 Data. Raw 210-Pb data.** File includes raw data used in geochronological modelling.
(XLSX)

## Acknowledgments

Additional acknowledgement goes to Richard Mahoney who, as a technician, assisted with all field days and much of the data collection for this work.

## Author Contributions

**Conceptualization:** Antonio B. Rodriguez, F. Joel Fodrie.

**Data curation:** Molly C. Bost, Charles D. Deaton, Antonio B. Rodriguez, Carson B. Miller.

**Formal analysis:** Molly C. Bost, Charles D. Deaton, Antonio B. Rodriguez, Brent A. McKee.

**Funding acquisition:** Antonio B. Rodriguez, F. Joel Fodrie.

**Investigation:** Molly C. Bost, Charles D. Deaton, Brent A. McKee.

**Methodology:** Molly C. Bost, Antonio B. Rodriguez.

**Supervision:** Charles D. Deaton.

**Validation:** Molly C. Bost, Charles D. Deaton, Brent A. McKee.

**Writing – original draft:** Molly C. Bost, Antonio B. Rodriguez.

**Writing – review & editing:** Molly C. Bost, Charles D. Deaton, Antonio B. Rodriguez, Brent A. McKee, F. Joel Fodrie, Carson B. Miller.

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
