## [Decision Letter · Decision Letter 0]

4 Oct 2022

PONE-D-22-15917Anthropogenic impacts on tidal creek sedimentation since 1900PLOS ONE

Dear Dr. Bost,

Thank you for submitting your manuscript to PLOS ONE. After careful consideration, we feel that it has merit but does not fully meet PLOS ONE’s publication criteria as it currently stands. Therefore, we invite you to submit a revised version of the manuscript that addresses the points raised during the review process.

In particular, both reviewers raised concerns (but also provided useful suggestions) regarding the structure of the manuscript. I agree with reviewer 1 that it would be beneficial to the manuscript if the research question could emerge more clearly and if the general flow of the text could be improved. Reviewer 1 provides a commented version of your original submission which, I believe, could prove useful while drafting a revised version.I also believe the three main points highlighted by reviewer 2 should each be addressed carefully. 

We look forward to receiving your revised manuscript.

Kind regards,

Goulven G Laruelle

Academic Editor

PLOS ONE

Journal Requirements:

2. In your Methods section, please provide additional information regarding the permits you obtained to collect samples for the present study. Please ensure you have included the full name of the authority that approved the field site access and, if no permits were required, a brief statement explaining why.

Reviewers' comments:

Reviewer's Responses to Questions

**Comments to the Author**

1. Is the manuscript technically sound, and do the data support the conclusions?

Reviewer #1: Yes

Reviewer #2: Partly

2. Has the statistical analysis been performed appropriately and rigorously? 

Reviewer #1: N/A

Reviewer #2: No

3. Have the authors made all data underlying the findings in their manuscript fully available?

Reviewer #1: Yes

Reviewer #2: Yes

4. Is the manuscript presented in an intelligible fashion and written in standard English?

Reviewer #1: Yes

Reviewer #2: Yes

5. Review Comments to the Author

Reviewer #1: There are some interesting observations in this paper and its is a useful case study of small watershed and the implications of land cover changes. However, those findings are masked by an extremely complex typology of the landscape which is very difficult to follow, not well explained, and seems to come out of the blue. The paper really needs a better set up with some clear research questions (grounded in this typology is appropriate) and much more on the regional setting. There are a large number of very general statements and its not clear if these are meant to be global observations or of regional/local interest only. I would focus the paper much more on the land cover and sediment with only enough coastal plain geomorphology to understand what is going on. For a global reach journal context for NC will be important. The attached marked up copy includes some specific comments that should be useful in reframing

Reviewer #2: This paper quantifies sediment accumulation rates in tidal creeks before and after 1950, identifies periods of greatest land cover change, and relates the changes in sedimentation to land use change in the creek watersheds. The results are valuable to better understand changes in sedimentation processes (in the light of coastal wetlands facing sea-level rise), and for coastal zone managers to avoid potentially undesired habitat transitions.

However, I have three main issues with the analysis and reporting of the results in this study.

First, the authors state that the timing of the increase in mass accumulation rates after 1959 corresponded with the major land cover change in 8 out of 12 creek sites. I understand that relating changes in mass accumulation rates with changes in land cover in a quantitative way is difficult because of different temporal resolutions (as the authors state in L444 – 447). However, how exactly do you define the time where mass accumulation rates start to increase? This is not so easy because in some cases you could argue that the increasing trend was already there before the major land cover shift (for instance, in creek 12 in Figure 6, or creek 11 where the increase seems to start later, around 1982, as is also stated in L462).

Is it possible to identify the trends and shifts in mass accumulation rates in a more quantitative way? For instance, piecewise regression could be used to show in a more objective way that the data follow different trends over time and identify that breakpoint.

Second, the authors only measured sedimentation rates at the bottom of creeks, but the resilience of salt marshes would largely depend on the deposition of sediment on the salt marsh platform. How would these results on the filling-up of creeks translate to sediment transport on the vegetated platform? In the end, salt marsh platforms need to accrete to prevent drowning with sea-level rise.

Finally, I think the manuscript structure should be improved. The Discussion section is mixed with a description of results and reference to figures that are introduced there for the first time. Figure 6 and 7 are only mentioned/described in the discussion: as far as I know, this is not very common (unless it’s a conceptual figure). I would first expect that all the main findings are presented in the results section (e.g., those reported in paragraph ‘Average SAR pre and post MLCC’).

Specific comments

L21-22: I’m not sure about the phrasing “Small coastal watersheds (tidal creeks)”: the watershed is the area drained by the tidal channel, but not the channel itself. Please rephrase.

L40: this sentence refers to ‘one site’ and ‘another site’, but it is unclear if it refers to a creek or one of the two main regions that you studied. I suggest making it more specific.

L211: What is the horizontal resolution of the LiDAR data? Is it the same for both sites?

Figure 4 legend: please indicate the difference between A and B plots in the legend: CC creek sites in A, and NHC creek sites in B?

Figure 5: maybe I missed this in the text, but can you explain what is causing this decline over the last 5 years or so?

L507: what is meant by structured habitats?

L535 and L585: is it 11 or 12 creeks in total? Creek 10 is sometimes missing in the figures.

L562: Fig should be Fig.

L583-585: that sounds like an overstatement, because the time of MAR acceleration is not quantified in an objective way.

6. PLOS authors have the option to publish the peer review history of their article (what does this mean?). If published, this will include your full peer review and any attached files.

Reviewer #1: No

Reviewer #2: No

---

## [Author Response · Author response to Decision Letter 0]

23 Nov 2022

I greatly appreciate the time and effort the both reviewers took in reviewing this manuscript. The suggestions were very constructive and helped us improve the manuscript immensely. We rearranged and removed a lot of the introduction and background as per the reviewers' suggestions in order to make the manuscript more concise. We think this is an important study and the changes we made that were suggested by the reviewers made it a strong paper.

---

## [Decision Letter · Decision Letter 1]

2 Jan 2023

Anthropogenic impacts on tidal creek sedimentation since 1900

PONE-D-22-15917R1

Dear Dr. Bost,

We’re pleased to inform you that your manuscript has been judged scientifically suitable for publication and will be formally accepted for publication once it meets all outstanding technical requirements.

Kind regards,

Goulven G Laruelle

Academic Editor

PLOS ONE

Additional Editor Comments (optional):

Dear authors,

I happy, as this year comes to a close, to inform you that both reviewers were satisfied with the modifications you made to your manuscript and were convinced by your answers to their comments. This means that your manuscript is accepted for publication in Plos ONE. As I will not be involved in the final steps towards the production of the final PDF, I would just like to point out that reviewer 2 noticed that there were some inconsistencies between the final cleaned up version of your manuscript and the one with the 'track changes' option. Similarly, figure 1 was missing from the last PDF. So, please, pay attention to these details when you upload the final files for the production of your article.

With that said, congratulations on your paper being accepted and all the best for 2023.

Reviewers' comments:

Reviewer's Responses to Questions

**Comments to the Author**

1. If the authors have adequately addressed your comments raised in a previous round of review and you feel that this manuscript is now acceptable for publication, you may indicate that here to bypass the “Comments to the Author” section, enter your conflict of interest statement in the “Confidential to Editor” section, and submit your "Accept" recommendation.

Reviewer #1: All comments have been addressed

Reviewer #2: (No Response)

2. Is the manuscript technically sound, and do the data support the conclusions?

Reviewer #1: Yes

Reviewer #2: Yes

3. Has the statistical analysis been performed appropriately and rigorously? 

Reviewer #1: N/A

Reviewer #2: Yes

4. Have the authors made all data underlying the findings in their manuscript fully available?

Reviewer #1: (No Response)

Reviewer #2: Yes

5. Is the manuscript presented in an intelligible fashion and written in standard English?

Reviewer #1: Yes

Reviewer #2: Yes

6. Review Comments to the Author

Reviewer #1: This paper has been appropriately revised and is now much clearer. Figure 1 howvere appears to be missing

Reviewer #2: Thank you for considering my comments. All my previous comments have been addressed, but I noticed that not all changes made in the track changes version have been accepted in the clean, unmarked version (for instance, this is the case for the sentence you added to the legend for Figure 4). Please make sure that all changes are incorporated.

7. PLOS authors have the option to publish the peer review history of their article (what does this mean?). If published, this will include your full peer review and any attached files.

Reviewer #1: No

Reviewer #2: No

---

## [Editor Report · Acceptance letter]

6 Jan 2023

PONE-D-22-15917R1 

Anthropogenic impacts on tidal creek sedimentation since 1900 

Dear Dr. Bost:

I'm pleased to inform you that your manuscript has been deemed suitable for publication in PLOS ONE. Congratulations! Your manuscript is now with our production department. 

Kind regards, 

on behalf of

Dr. Goulven G Laruelle 

Academic Editor

PLOS ONE